



High-resolution observation of stable carbon isotope ratios of water-soluble organic
carbon in particle/gas phases at an urban site in China: Using an improved isotope
ratio mass spectrometry method
Hao-Ran Yu, Yan-Lin Zhang*, Fang Cao, Xiao-Ying Yang, Tian Xie, Yu-Xian Zhang, Yongwen
Xue
[1] School of Applied Meteorology, Nanjing University of Information Science & Technology,
Nanjing 210044, China.
[2] Atmospheric Environment Center, Joint Laboratory for International Cooperation on Climate and
Environmental Change, Ministry of Education (ILCEC), Nanjing University of Information
Science & Technology, Nanjing 210044, China.
**TOC:**

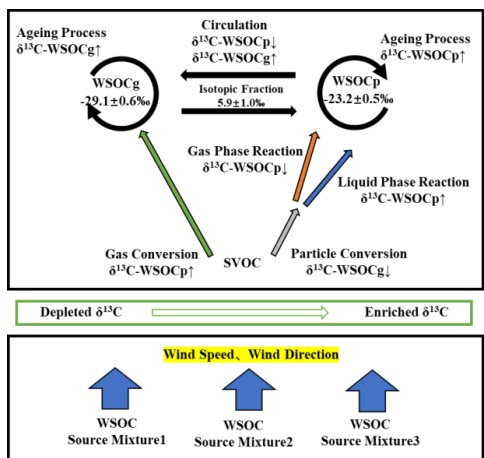

**Key points:**
1. Obvious fractionation exists between $\delta^{13}$C-WSOCp and $\delta^{13}$C-WSOCg, and there is obvious

15       diurnal variation of it.

2. Mutual conversion between WSOCp and WSOCg only happens during daytime, but radiation

17       intensity has no effect on the formation mechanism and gas-particle distribution mechanism of

18       WSOC.

3. Gas-particle distribution mechanism of WSOC was restricted by meteorological conditions,

20       precursor gaseous pollutants, gaseous oxidants and acid gases, and is accompanied by the

21       formation of secondary inorganic ion component





**Abstract**: A high time resolution synchronous sampling method along with determination of stable
carbon isotopes of gaseous water-soluble organic carbon (WSOCg) and particulate water-soluble
organic carbon (WSOCp) was realized in this research through equipment modification and method
improvement. It was found that WSOCg has significant higher concentration than WSOCp and a
more depleted $\delta^{13}$C. Both concentrations of WSOCp and WSOCg have the seasonal variation
characteristics of high in winter (WSOCp=15.7±3.9μg/m³; WSOCg=42.4±6.0μg/m³) and low in
summer (WSOCp=5.9±1.8μg/m³; WSOCg=25.2±5.2μg/m³), with greater increase in WSOCp
(167%) than that in WSOCg (68%). During wintertime, WSOCp and WSOCg had similar daily
variation characteristics of concentration, and opposite daily variation characteristics of $\delta^{13}$C.
WSOCp had a bimodal distribution with obvious low value at sunrise and sunset, while $\delta^{13}$C-
WSOCp had a unimodal distribution with low in daytime (-24.6±1.1‰) and high in nighttime (-
22.3±1.7‰). WSOCg and $\delta^{13}$C-WSOCg had same distribution with high in daytime and
(49.3±8.8μg/m³; -27.9±1.1‰) low in nighttime (38.3±4.6μg/m³; -29.9±0.4‰). Combining the $\delta^{13}$C
variation characteristics with the synchronous observation results of meteorological conditions,
gaseous precursor pollutants, gaseous oxidants, gaseous acids and fine particle components, the
restriction factors of WSOC gas-particle distribution mechanism were discussed. The presence of
radiation rather than its intensity decided whether generations process of WSOCp and WSOCg are
divided, for $\delta^{13}$C-WSOC of two phases showed significant correlation only during daytime.
Meteorological conditions, gaseous precursor pollutants, gaseous oxidants and gaseous acids restrict
the gas particle distribution of WSOC by affecting the aging process of WSOCp and WSOCg, gas-
particle conversion ratio of semi-volatile organic compounds (SVOC) and the gas phase and liquid
phase generation ratio of WSOCp. At the same time, the gas-particle distribution process of WSOC
is strongly related to the formation of secondary inorganic ions (nitrate, sulfate, ammonium), and
the gas-particle distribution between gaseous nitrous acid and nitrite.
**Keywords**: gas phase, particle phase, WSOC, $\delta^{13}$C.
1.  Introduction
Carbonaceous aerosols were considered to be one of the largest sources of uncertainty in the
estimation of global radiative forcing in climate change research (Bond, et al., 2013; Pöschl, 2005),
and water soluble organic carbon is an important component of carbonaceous aerosols, which has
an important impact on haze formation, human health and the earth's radiation balance. Generally,



carbonaceous aerosols include organic carbon (OC), elemental carbon (EC) and carbonate carbon
(CC) aerosols. For the content of CC is generally low (<5%) and its properties are relatively stable,
research of atmospheric environment and climate mainly focuses on EC and OC, which can account
for 20-50% of the mass concentration of $PM_{2.5}$(Pöschl, 2005). According to the water solubility of
OC, it can be divided into water insoluble organic carbon (WIOC) and water-soluble organic carbon
(WSOC). Most of WSOC comes from biomass burning or secondary organic aerosol (SOA)(Kondo,
et al., 2007; Weber, et al., 2007), mainly contains dicarboxylic acids, ketoacids, dicarbonyl
compounds, long-chain fatty acids and short-chain monocarboxylic acids, which can contribute 20-
80% of OC mass concentration. However, there exist quite different distribution characteristics of
WSOC in different regions(Fu, et al., 2013; Ho, et al., 2006; Miyazaki, et al., 2006; Wang, et al.,

63  2012).

The formation of WSOC is closely related to gas-particle distribution, emission source and
meteorological conditions. The synchronous study on particle phase of WSOC (WSOCp) and gas
phase of WSOC (WSOCg) is helpful to understand the sources and atmospheric process of SOA.
Though WSOCp was considered to characterize SOA to some extent when the direct contribution
of biomass combustion can be neglected(Kondo, et al., 2007; Weber, et al., 2007),more and more
evidences showed an important contribution of WSOCg to the generation of SOA.。Similar to
WSOCp, WSOCg can be generated from secondary conversion of VOCs, in addition to the direct
emission such as biomass burning and fossil fuel combustion(Carlton, et al., 2009; Liu, et al., 2012;
Liu, et al., 2012; Meng, et al., 2014). There is evidence that VOCs emitted from natural or man-
made sources can be oxidized to WSOCg in the atmosphere (Carlton and Turpin, 2013; Sareen, et
al., 2017), and liquid phase reaction of WSOCg absorbed by cloud water or aerosol liquid water is
one of the important ways to formation of SOA(Liu, et al., 2012). In addition, a large part of the
intermediate products of isoprene oxidation, which was considered to be the most important
precursor of SOA, can constitute WSOCg, including glyoxal, methylglyoxal, and isoprene
epoxydiol, and some low molecular organic acids (such as formic acid and acetic acid) (Carlton, et
al., 2009; El-Sayed, et al., 2015). However, there is still great uncertainty in assessing the
contribution of natural and man-made sources to WSOCg under the background of compound
atmospheric pollution. Only qualitative or semi quantitative calculation can be carried out by means
of characteristic molecular markers, while the results based on model simulation are limited by the
understanding of atmospheric mechanism and lack of direct observation evidence for verification.
Previous researches had studied on gas-particle distribution mechanism of WSOC by
synchronously measuring the concentration of WSOCp and WSOCg. The research in Atlanta
reported that the WSOCg in summer ranged from 1.1 to 73.1 µgC m$^{-3}$, average 13.7µgC m$^{-3}$, which
was obviously higher than WSOCp (3.3±1.8µgC m$^{-3}$), but had a certain linear correlation with
WSOCp (Hennigan, et al., 2009)。Later,research in Atlanta and Los Angeles found that WSOCp
and WSOCg had similar diurnal variation rules, and both reached the highest value after noon (about
14:00), .indicating important contribution of secondary reaction(Zhang, et al., 2012). It was also
found that difference of precursors may lead to the change of WSOC in the gas-particle distribution
relationship. The variation of WSOCp in Atlanta was controlled by the aerosol liquid water content
(ALWC) under the influence of humidity, which was similar to WSOCp in Chongming island,
China(Lv, et al., 2022). While variation of WSOCp in Los Angeles was independent of ALWC, but
depended on OC concentration(Zhang, et al., 2012). Similarly, variation of WSOCp in Prague was
found to depend on TC in gas phase. There may be obvious regional differences in the gas-particle
distribution relationship of WSOC.
Stable carbon isotope ratio ($^{13}$C/$^{12}$C, $\delta^{13}$C) can provide important information about the sources
and atmospheric chemical conversion processes of carbonaceous aerosols. $\delta^{13}$C was generally used
to distinguish sources of carbonaceous aerosols, such as C3 and C4 plant, vehicle exhaust, coal
combustion and other fossil sources(Cao, et al., 2011). However, $\delta^{13}$C has a mass dependent isotope
fractionation phenomenon, leading an isotope variation apart from effects of sources. The secondary
reaction of VOCs to generate SOA is an important source of WSOCp(Kondo, et al., 2007; Weber,
et al., 2007), and aerosols formed by VOCs usually have a more depleted $\delta^{13}$C than the
precursor(Anderson, et al., 2004; Rudolph, et al., 2003; Rudolph, et al., 2000). For example, isotope
fractionation in the process of biosynthesis of isoprene will lead to $\delta^{13}$C of isoprene 2.6 ± 0.9 ‰
smaller than it in blade(Rudolph, et al., 2003). The main scavenging pathway of VOCs is its reaction
with OH radical and ozone, and these atmospheric oxidants tend to react with VOCs depleted $\delta^{13}$C
(reverse dynamic isotope effect), resulting in the $\delta^{13}$C enrichment of residual VOCs in the
atmosphere and $\delta^{13}$C dilution of particles as oxidation products(Anderson, et al., 2004; Rudolph, et
al., 2003; Rudolph, et al., 2000). Some studies have also shown that $\delta^{13}$C will be enriched in particles
during ageing process, such as process in which binary acid reacts with OH and is removed in the



form of $CO_2$/CO(Aggarwal and Kawamura, 2008; Noziere, et al., 2015; Pavuluri, et al., 2011; Wang,
et al., 2012; Zhang, et al., 2016). Measurement based on high time resolution observation of $\delta^{13}C$-
WSOCp and $\delta^{13}C$-WSOCg will provide a new perspective for deep understanding of the whole life
cycle of WSOC.

2.   Data and methods
2.1 Sampling and chemical analysis
The two phases WSOC samples were collected from 4 to 14 December of 2021 in Nanjing, a
megacity in eastern China. The sampling site is located at the agrometeorological station of
Nanjing University of Information Science and Technology, which is close to a busy traffic road
and surrounded by a large number of industrial factories. The WSOC mass concentration was
measured with the total organic carbon analyzer (TOC-L, Shimadzu, Japan), analyzed along with
peak area signal of isotope ratio mass spectrometer (MAT253, Thermo Fisher Scientific, USA).
Six anions ($F^-$, $Cl^-$, $NO_2^-$, $NO_3^-$, $SO_4^{2-}$ and $PO_4^{3-}$), five cations ($Na^+$, $NH_4^+$, $K^+$, $Mg^{2+}$, and $Ca^{2+}$)
and three organic acids (formic, acetic, and oxalic acids) were measured using the ion
chromatography system (IC-5000, Thermo Fisher Scientific, USA). The $PM_{2.5}$ and pollution gas
concentration were observed at Pukou environmental supervising station, and the meteorological
data was observed at Pukou meteorological station. The radiation data are from the shared data of
the comprehensive meteorological observation base of Nanjing University of Information Science
and Technology. During 3 June and 13 July of 2021, 202 samples were collected with ions and
WSOC concentration analyzed, which was used as summer data compared with winter mass
concentration.
A wet annular denuder (WAD) combined with scrub and impactor aerosol collector (SCI)
(IGAC, Machineshop, Taiwan, China) was used for water-soluble organic carbon collecting. .Based
on the principle of gas diffusion, gas phase of WSOC was absorbed and washed away by the
absorption liquid flowing through the pipe wall of WAD. The steam generated by the SCI will
capture the particle phase of WSOC by collision and condensation after that, and the sample will be
collected by inertial impact in liquid form. A very sharp cut cyclone (VSCC) was set in the entrance
of to filter particles above 2.5μm. WSOCp and WSOCg samples were collected in glass tubes by an
automatic rotating device at the end of system. The system can collect 20mL WSOCp and 20mL
WSOCg samples from 1 m³ of air every hour. The system pipeline should be flushed with ultrapure
water for one week before each sampling to reduce blank pollution, and the glass tubes were
prebaked at 450°C for 6 h before use. Blank pollution of WSOC accounted 8% of average WSOCp
and 3% of average WSOCg. To analyze filter-based WSOCp, $PM_{2.5}$ aerosol samples were also
collected onto quartz fiber filters which were prebaked at 450 °C for 6 h using an automatic sampler
(DHA-80, Digital, Switzerland). Sampling flow rate was 500L/min and the time resolution was 1 h.
Daily average of filter-based WSOCp accounted for 81% of WSOCp determined by IGAC system.
It was similar to a percentage of 89% in a study using IGAC system at Chongming island(Anderson,
et al., 2008). It may accounted to the speciation of the sampled organic gases which resulted in the
variation in denuder efficiency(Anderson, et al., 2008). However, the lower WSOCp sampling
efficiency of denuder compared with filter is still hard to explain, for evaporation loss of OC,
especially volatile and semi volatile carbonaceous components, was proved to happen on filter
surfaces(Lewtas, et al., 2001; Yang, et al., 2021). It is considered that the denuder had passive
sampling capability of WSOC, but it's hard to estimate the artifacts accurately for there was a
variation of passive sampling along with pollution level(Jingyue, et al., 2010).

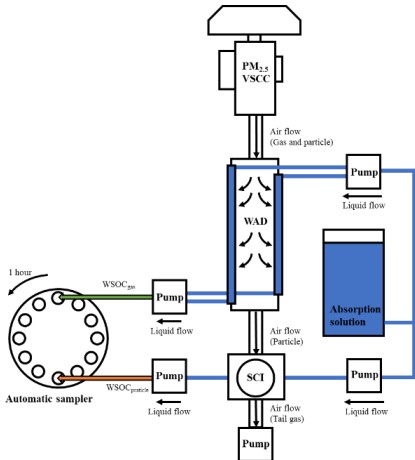


Fig 1. High resolution WSOCp and WSOCg synchronous sampling device


2.2  Stable carbon isotope analysis

δ¹³C-WSOC values were analyzed with a GasbenchII-IRMS system. Due to the low content of

WSOC in liquid samples, an improved method was used for determination of δ¹³C in this study,



mainly included pretreatment improvement and instrument transformation. Subsequent data
analysis used person correlation analysis.
2.2.1    Sample pretreatment and isotope determination
The wet oxidation method was used to pretreat the samples in order to convert WSOC to $CO_2$。
It took 4mL filtered liquid sample (0.22μm, Hydrophobic PTFE needle filter), along with 1mL
acidifier (100μL 85% phosphoric acid, soluble in 100mL Milli-Q water), standing for 1h in order to
remove WSIC ($HCO_3^-$, $CO_3^{2-}$). Otherwise, WSIC may cause a isotope fractionation of about
0.2 ‰(Suto and Kawashima, 2018). Oxidizer was prepared by 1g potassium persulfate ($K_2S_2O_8$,
99.99%), 50μL of phosphoric acid ($H_3PO_4$, 85%) and 100mL of Milli-Q water. It took 0.5mL of
oxidizer injected into a 12mL pre-purged sealed headspace bottle (100mL/min high purity helium
purging for 1 min). 2mL liquid sample was then injected into headspace bottle, and heated in sand
bath with 100℃ for 1h. It took 6h to wait for complete condensation of water vapor after sand bath
to avoid leakage of headspace bottle or damage of IRMS.
$CO_2$ blank in headspace bottle mainly came from three parts, OC of oxidizer/acidifier(Fisseha,
et al., 2006), $CO_2$ dissolved in Milli-Q water and residual air $CO_2$ during purging. During the $CO_2$
blank test, the signal response of the mass spectrometer to samples was controlled to be 80 area/μgC.
Compared with helium purging under liquid level after sample injection, a method of helium purging
before sample injection could effectively eliminate the impact of residual CO2 in the air (<2area).
In addition, $CO_2$ blank signal would increase around 10area (about 0.1μgC) with each 50mg increase
of $K_2S_2O_8$. And $CO_2$ blank signal would increase around 20area (about 0.25μgC) with each 1mL
increase of liquid in headspace bottle. In contrast, $CO_2$ blank is more affected by liquid content. And
this part of $CO_2$ blank couldn't be controlled by He purging or pre-heating. By controlling the
sample volume, reducing the oxidant concentration and injection volume, the total $CO_2$ blank signal
finally reached around 30area (about 0.3μgC), approximately 19% of the average carbon content of
WSOC sample.
The Gasbench-IRMS system and determination method were improved in this method. The
system used high-purity helium as carrier gas. Sample gas was pushed through the water trap
(magnesium perchlorate) and VOC trap in the preconcentration unit (Precon) by helium at a pressure
of 1.7bar. After 260s of freeze enrichment and impurity removal in liquid nitrogen trap, helium at a
pressure of 0.6bar was switched by rotating of six-way valve in Precon, pushing sample gas into gas



chromatographic column (Polra PLOT Q) to separate $N_2O$ and $CO_2$. Back purge valve of front
pipeline was open at the same time to purge the sample injection pipeline. Finally, sample gas
entered IRMS for $\delta^{13}C$ determination after water removal through a Nafion permeation tube. It took
a total of 24min in this method, and the determination accuracy can reach 0.25 ‰ above 1μgC, can
reach 0.53 ‰ above 0.5μgC.

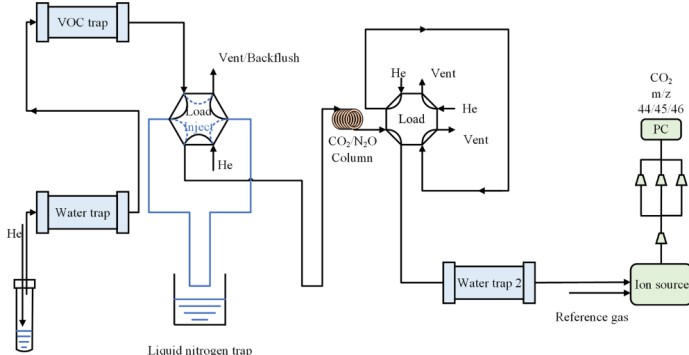


Fig 2. Stable carbon isotope ratio determination system

3.  Results and discussion
3.1  Improvement of determination method
Determination limit was improved to less than 5μgC in this study(Zhang, et al., 2019).
However, determination values have obvious peak area independence in this range of carbon content
(Fig. 3a). Based on the consideration of drug solubility in water, three working standards were used
in this study in order to establish the standard curve between true value and determination value
(Fig. 3b): potassium hydrogen phthalate (KHP) and two kinds of sucrose (Suc-1 and Suc-2) with
different $\delta^{13}C$. The carbon isotope composition of these three standards is analyzed by combustion
method, using an elemental analyzer combined with an isotope ratio mass spectrometer (EA-IRMS,
Thermo Fisher Scientific, USA), as follows: -12.08‰ (Suc-1), -24.83‰ (Suc-2), and -30.62‰
(KHP). This range of $\delta^{13}C$ values is able to cover the majority of the $\delta^{13}C$-WSOC values in ambient
air samples. Standards were resolved in Milli-Q water (resistivity 18.2MΩ) to make standard
solutions of a carbon content of 0.5, 1, 2 and 4 μg in 2mL standard solution to test the procedures
during the pretreatment. What's more, it was found in this study that over-heated oxidant would
cause severe dilution of $\delta^{13}C$ in standard samples.





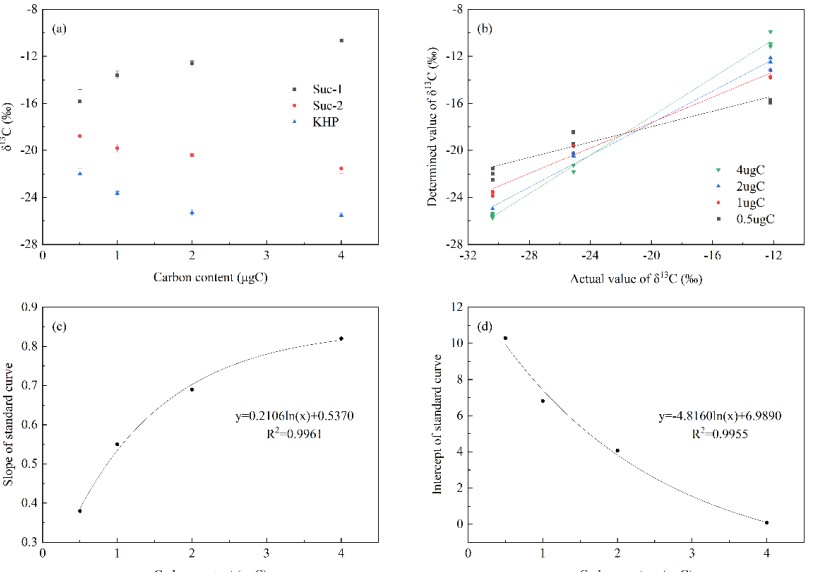


Fig 3. Determination value of three isotopic working standard with different carbon contents. (a)
Peak area dependent effect of $\delta^{13}C$; (b) Standard curve with different carbon content; (c) Slope
correction curve of standard curve; (d) Intercept correction curve of standard curve.

3.2  Component and concentration variation
Table 1 Concentration distribution of different components

| $\mu g/m^3$ | WSOC | WSII | $PM_{2.5}$ | Formate | Acetate | Oxalate |
|---|---|---|---|---|---|---|
| Particle phase | 15.7±3.9 | 29.1±13.6 | 57.0±26.9 | 0.16±0.07 | 0.30±0.20 | 0.18±0.03 |
| Gas phase | 42.4±6.0 | 25.1±10.9 | | 0.54±0.50 | 5.66±3.23 | 0.16±0.03 |


During study period, WSOCp was in the range of 8.7~35.8 µg C m$^{-3}$ and had an average
concentration of 15.7±3.9 µg C m$^{-3}$. WSOCp accounted for 27.5% of the mass concentration of
$PM_{2.5}$, and there was a significant correlation between them ($p<0.01$). The range of WSOCg
concentrations observed was 25.6~61.2 µg C m$^{-3}$, and the mean concentration was 42.4±6.0 µg C
m$^{-3}$. Overall, there is a very significant level of positive correlation between WSOCp and WSOCg
($p<0.01$), which indicates a common source of two phases of WSOC. WSOCp and WSOCg were
both positively correlated with CO, $PM_{2.5}$ and $PM_{10}$ ($p<0.01$), indicating a common CO emission
source of WSOCp and WSOCg and the significant contribution of WSOCg to sever haze





pollution. It seems concentration of WSOCg was not affected by meteorological condition in
winter, but relative humidity could enhance generation of WSOCp.
WSOCp and WSOCg were obvious higher in wintertime. During summertime, WSOCp was
in the range of 1.7~12.1 µg C m$^{-3}$ and had an average concentration of 5.9±1.8 µg C m$^{-3}$. The
range of WSOCg concentrations observed was 16.4~40.0 µg C m$^{-3}$, and the mean concentration
was 25.2±5.2 µg C m$^{-3}$. Enhancement of WSOCp (166%) was stronger than WSOCg (68%),
indicating a more favorable environment for gas phase to particle phase transformation in
wintertime.
Diurnal profiles of WSOCp and WSOCg were similar, indicating importance of
photochemical reaction in WSOC formation (Fig4a, Fig4b). Both WSOCp and WSOCg reached
daily maximum concentration at early noon (at 10:00 LT), and reached relatively low
concentrations at early morning (about 6:00 LT) and early evening (about 18:00 LT). Differently,
there were obvious valley concentration of WSOCp appeared at sunrise and sunset (at 07:00 LT
and 17:00 LT), indicating different generation process of WSOCp in daytime and night time.
Compared with WSOCp (55%), WSOCg had a relatively lower enhancement (28%) after sunrise.
It suggests a substantial regional background and a relatively long lifetime of WSOCg.

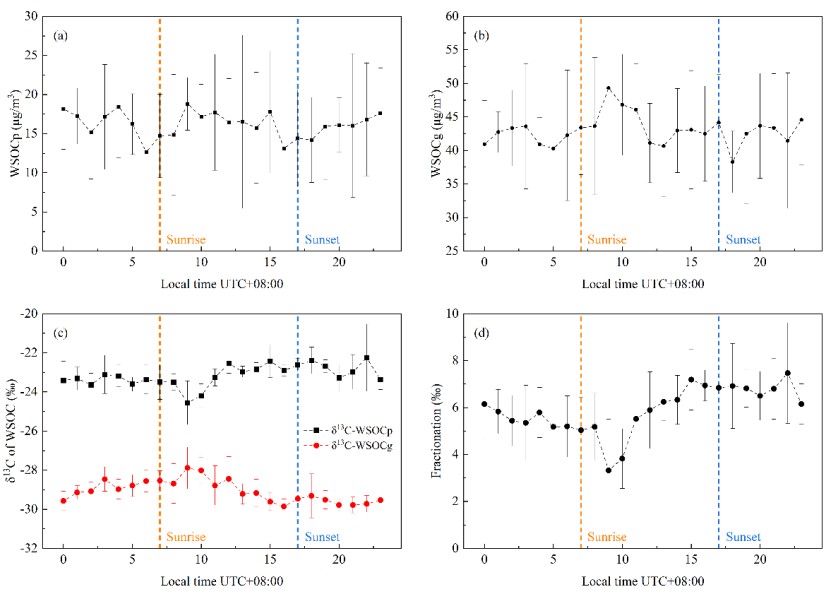


Fig. 4 Diurnal variation distribution (a) Concentration of WSOCp; (b) Concentration of WSOCg;
(c) $\delta^{13}C$ of WSOCp and WSOCg; (d) $\delta^{13}C$ fractionation between WSOCp and WSOCg.

Different from similar concentrations of water-soluble inorganic ions (WSII) in two phases,
WSOCg was apparently higher than WSOCp, and components of WSOC in different phase may
also differed greatly. As a typical component of WSOC, organic acids mainly existed in gas phase.
Acetic acid, formic acid and oxalic acid were the main organic acids in WSOC, and the fraction
ratios in the particle phase were 0.05, 0.23 and 0.53 respectively.

3.3  Stable carbon isotopic variation characteristic
WSOCp had a more enriched $\delta^{13}C$ compared with WSOCg. During study period, $\delta^{13}C$-WSOCp
was in the range of -22.5~-28.0 ‰ and had an average $\delta^{13}C$ of -25.9±0.7 ‰, $\delta^{13}C$-WSOCg was in
the range of -24.3~-31.5 ‰ and had an average $\delta^{13}C$ of -29.9±0.9 ‰. Both $\delta^{13}C$-WSOCp and $\delta^{13}C$-
WSOCg had an obvious diurnal variation characteristic. $\delta^{13}C$-WSOCp was high in night time and
low in day time, and $\delta^{13}C$-WSOCg was on the contrary. Difference between $\delta^{13}C$-WSOCp and $\delta^{13}C$-
WSOCg reached minimum at around 09:00 LT, and reached maximum at about 23:00 LT (Fig4c).
Difference between $\delta^{13}C$-WSOCp and $\delta^{13}C$-WSOCg, which was considered as isotope
fractionation between two phases of WSOC, had negative correlation with concentration of CO、
$PM_{2.5}$、$PM_{10}$、WSOCg and relative humidity (RH) above significant level ($p<0.05$), had positive
correlation with $O_3$ on an extremely significant level in a daily view. However, only $O_3$, WSOCg,
$PM_{2.5}$, $PM_{10}$ had extremely significant correlation with $\delta^{13}C$-WSOCp, $\delta^{13}C$-WSOCg and
fractionation at the same time ($p<0.01$). For there was extremely significant positive correlation
between fractionation and WSOCg, $PM_{2.5}$ and $PM_{10}$, main source of WSOC and its emission
intensity may have an obvious diurnal change (Such as liquid fossil-fuel, $\delta^{13}C$ of which is about -
25.5±1.3‰(August, et al., 2015)).
After distinguishing daytime and nighttime according to radiation intensity, it was found that
decrease of temperature and $O_3$, along with increase of WSOCg and RH, promoted decreasing of
fractionation on an extremely significant level during daytime, and increase of $O_3$, $PM_{2.5}$ and $PM_{10}$
promoted increase of fractionation on an extremely significant level during nighttime ($p<0.01$).
Though there was no correlation between fractionation and radiation, there was not only a negative
correlation on an extremely significant level ($p<0.01$), but also a highly fitting linear response



relationship ($R^2$=0.63) between $\delta^{13}$C-WSOCp and $\delta^{13}$C-WSOCg in an environment with non-zero
radiation (Fig5a). It indicated that the mutual conversion between WSOCp and WSOCg may only
exists during daytime, and the generations of WSOCp and WSOCg were two independent processes.
We also found it was interesting that fractionation had a significant negative correlation with the
solar radiation after 3 hours ($p<0.05$), along with a highly fitting linear response relationship
($R^2$=0.51), which was hard to explain (Fig5b).

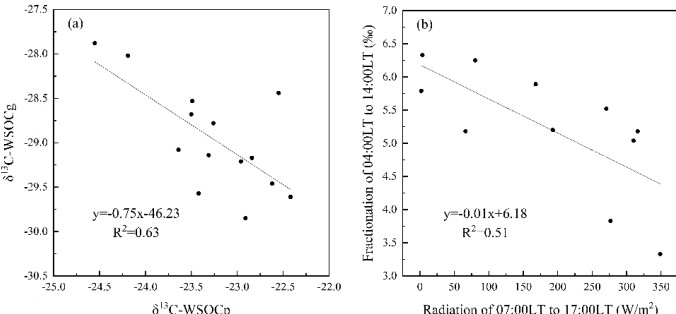


Fig. 5 Linear response relationship during daytime (a) $\delta^{13}$C-WSOCp and $\delta^{13}$C-WSOCg; (b)

Fractionation (04:00LT to 14:00LT) and Radiation (07:00LT to 17:00LT).


Enrichment of $\delta^{13}$C-WSOCg had an extremely significant correlation with the increase of
WSOCp and WSOCg at the same time ($p<0.01$). However, dilution of $\delta^{13}$C-WSOCp only had a
significant correlation with the increase of WSOCg ($p<0.05$), with no correlation with WSOCp. It
was believed that increase of WSOCp and WSOCg was attributed to changes in emission intensity.
WSOCg is more affected by the direct emission source, and WSOCp more from the conversion of
gaseous substances rather than direct emission source.
3.3 Effects of gaseous pollutants
Isotopic fraction of $\delta^{13}$C between gas phase and particle phase of WSOCp and WSOCg was
considered mainly to mainly come from five parts: isotope fractionation due to gas-particle
distribution of SVOC(Vodicka, et al., 2022); isotope fractionation due to equilibrium exchange of
WSOCp and WSOCg(Gensch, et al., 2014); isotope changes caused by source proportion changes
and emission intensity changes of WSOCp and WSOCg(Saehee, et al., 2020); enrichment of $\delta^{13}$C
caused by retention and aging of WSOCp and WSOCg; isotope fractionation due to proportion





change of gas phase reaction and liquid phase reaction in generation process of WSOCp. According
to the principle of isotope mass conservation showed as formula (F1), with a certain mass of SVOC,
more gas phase conversion will result in enrichment of $\delta^{13}$C-WSOCp, and more particle phase
conversion will result in dilution of $\delta^{13}$C-WSOCg。 The conversion of WSOCp and the dissolution
of WSOCg on the aerosol surface may result in consistent $\delta^{13}$C of WSOCp and WSOCg due to
isotope exchange. Changes in the sources of WSOCp and WSOCg will directly cause $\delta^{13}$C-WSOCp
and $\delta^{13}$C-WSOCg to approach the $\delta^{13}$C of main source(Cao, et al., 2011), and concentration changes
of WSOCp and WSOCg only happens when emission intensity changes. It had been proved that
oxidants in atmosphere tend to react with VOCs which have more depleted $\delta^{13}$C (Anderson, et al.,
2004; Rudolph, et al., 2003; Rudolph, et al., 2000), leading to a more enriched $\delta^{13}$C in residual
VOCs. Aged OC in aerosol was also proved to have more enriched $\delta^{13}$C(Aggarwal and Kawamura,
2008; Pavuluri, et al., 2011; Wang, et al., 2012; Zhang, et al., 2016). Therefore, it's believed that
aging of WSOC leads to more enriched $\delta^{13}$C-WSOCp and $\delta^{13}$C-WSOCg. While there is a more
enriched $\delta^{13}$C in residual VOCs due to reverse kinetic isotope effect of oxidants(Noziere, et al.,
2015), SOA generated by gas phase reaction will have more enriched $\delta^{13}$C, which means $\delta^{13}$C of
SOA will be depleted when proportion of liquid phase reaction increases. Similarly, it was believed
increased proportion of gas phase reaction leaded to dilution of $\delta^{13}$C-WSOCp, increased proportion
of gas phase reaction leaded to enrichment of $\delta^{13}$C-WSOCp.
$$\frac{WSOC_p \times \delta^{13}C(WSOC_p) + WSOC_g \times \delta^{13}C(WSOC_g)}{WSOC_p + WSOC_g} = SVOC \times \delta^{13}C(SVOC) \quad (F1)$$

In which WSOCp and WSOCg are the parts converted from SVOC.
Increase of CO leaded to enrichment of $\delta^{13}$C-WSOCg on an extremely significant level
(p<0.01). Though there was also extremely significant positive correlation between concentration
of CO and WSOCp, WSOCg (p<0.01), along with significant positive correlation between
concentration of CO and fp-WSOC (p<0.05), it was difficult for CO to affect conversion between
WSOCp and WSOCg by participating in the atmospheric chemical reaction. For higher
concentration of CO indicated a stronger primary emission source(August, et al., 2015), it was
believed that emission of WSOC enhanced in an environment with high concentration of CO,
leading generation and retention of plenty of WSOCg, along with aging of those WSOCg. However,
there was no effect on gas-particle distribution mechanism of WSOC.



Increase of $NH_3$ leaded to dilution of $\delta^{13}C$-WSOCp, enrichment of $\delta^{13}C$-WSOCg and increase
of WSOCg on an extremely significant level ($p<0.01$). For there was research found increase of
ALWC along with decrease of pH of aerosol in environment of high concentration of $NH_3$(Shaojun,
et al., 2022), which may be conducive to aerosol release WSOCg. Dilution of WSOCp may came from
mutual conversion of WSOCp and WSOCg, while enrichment of $\delta^{13}C$-WSOCg may came from aging
of WSOCg which was generated in an environment of high concentration of $NH_3$ considering
increasing concentration of WSOCg. There was also possibility of increasing emission intensity of
specific source, which leaded to $\delta^{13}C$-WSOCp and $\delta^{13}C$-WSOCg tended to be consistent. However,
it was hard to explain why there was no significant effect on WSOCp. Considering both $\delta^{13}C$-
WSOCp and $\delta^{13}C$-WSOCg closed to -26‰ at the same time when fractionation decreased, which
is similar to characteristic $\delta^{13}C$ value of liquid fossil-fuel(August, et al., 2015), it was believed that
liquid fossil-fuel was a common source during a period of high concentration of $NH_3$.
With no correlation with concentration of neither WSOCp nor WSOCg, increase of $NO_2$ leaded
to dilution of $\delta^{13}C$-WSOCp and $\delta^{13}C$-WSOCg at the same time on an extremely significant level
($p<0.01$). For there is dilution of $\delta^{13}C$-WSOCp along with enrichment of $\delta^{13}C$-WSOCg at the same
time, there was little possibility of dilution of $\delta^{13}C$-WSOCp caused by mutual conversion of WSOCp
and WSOCg. Considering evidence for gas phase reaction of organic nitrates(Alexander, et al., 2019),
it was believed that SVOC may be more converted to particle phase, and the gas phase formation
reaction of SOA was strengthened under high concentration of $NO_2$.
Increase of $O_3$ leaded to enrichment of $\delta^{13}C$-WSOCp and dilution of $\delta^{13}C$-WSOCg on an
extremely significant level ($p<0.01$). Similar with $NO_2$, neither WSOCp nor WSOCg had correlation
with concentration of $O_3$, which means none variation of emission intensity of WSOC source when
$\delta^{13}C$ changed. It was believed that SVOC conversed more to WSOCp under high concentration of
$O_3$, leading to of dilution of $\delta^{13}C$-WSOCg. The aging of WSOCp was promoted at the same time,
leading to enrichment of $\delta^{13}C$-WSOCp. Dilution of $\delta^{13}C$-WSOCg caused by conversion from SVOC
to WSOCp exceeded enrichment of $\delta^{13}C$-WSOCg caused by aging of WSOCg, leading to a more
dilute $\delta^{13}C$-WSOCg, indicating that $O_3$ tends to react with SVOC rather than WSOCg.
It seems different gaseous acids had different effects on WSOC. Increase of HONO leaded to
dilution of $\delta^{13}C$-WSOCp and enrichment of $\delta^{13}C$-WSOCg over a significant level ($p<0.05$). For
there was also extremely significant positive correlation between HONO and WSOCp, WSOCg



(p<0.01), emission intensity of WSOC sources may increase under high concentration of HONO,
along with mutual conversion of WSOCp and WSOCg. Considering that HONO participates in
formation of SOA as an oxidant(Chi, et al., 2022), there was also possibility of enhanced gas phase
generation of WSOC leaded to dilution of $\delta^{13}$C-WSOCp. Increase of HCl leaded to enrichment of
$\delta^{13}$C-WSOCg and increase of WSOCg on a significant level (p<0.05), with no correlation with $\delta^{13}$C-
WSOCp. The presence of HCl may inhibit the transformation from WSOCg to WSOCp, promoting
the retention and aging of WSOCg. Differs from other acids, gaseous oxalic acid leaded to dilution
of $\delta^{13}$C-WSOCg on a significant level (p<0.05) without affecting concentration of WSOCp and
WSOCg, which may be caused by the increase in proportion of SVOC conversion to WSOCp.
3.4 Effects of component of $PM_{2.5}$ and gas-particle distribution
Increase of $PM_{2.5}$ and $PM_{10}$ leaded to enrichment of $\delta^{13}$C-WSOCg on an extremely significant
level (p<0.01). There was also extremely significant positive correlation between $PM_{2.5}$, $PM_{10}$ and
WSOCp, WSOCg. For there was no correlation between $PM_{2.5}$, $PM_{10}$ and $\delta^{13}$C-WSOCp, it was
believed that emission intensity of WSOC sources increased with none change of main source under
high concentration of $PM_{2.5}$ and $PM_{10}$. Aging of WSOCg and concentration increase of particle
matters had consistent atmospheric environmental conditions, and aging of WSOCg may make a
considerable contribution to formation of haze.
Different components of $PM_{2.5}$ had different responses to the concentration and isotopes of
WSOCp and WSOCg. Consistent with $PM_{2.5}$ and $PM_{10}$, $SO_4^{2-}$, $NO_3^-$ and $NH_4^+$ (SNA) all had
extremely positive correlation with enhancement of $\delta^{13}$C-WSOCg, along with increase of WSOCp
and WSOCg (p<0.01). Aging of WSOCg may cause generation of haze by participating in formation
of SNA. Apart from extremely positive correlation with WSOCp and WSOCg, increase of $SO_4^{2-}$
leaded to enrichment of $\delta^{13}$C-WSOCp on a significant level (p<0.05). For sulfate is largely formed
in liquid phases(Gehui, et al., 2018; Gehui, et al., 2016), it was believed that liquid phase generation
reaction of WSOCp enhanced during haze period with high concentration of $SO_4^{2-}$. Increase of $NO_2^-$
leaded to dilution of $\delta^{13}$C-WSOCp on an extremely significant level (p<0.01), indicating a possible
enhancement in gas phase generation reaction of WSOCp. However, formation of $NO_2^-$ in
atmosphere is still uncleared.
Generally, fp=WSOCp/(WSOCp+WSOCg) represented gas-particle ratio of WSOC(Hennigan, et
al., 2009). fp of ions was used to represent gas-particle ratio of ions in this research. It was found





that though increase of fp of SNA promoted increase of WSOCp and WSOCg in the same time on
a significant level ($p<0.05$), increase of fp-$NH_4^+$ only related to enrichment of $\delta^{13}$C-WSOCp and
increase of fp-$NO_3^-$ and fp-$SO_4^{2-}$ only related to enrichment of $\delta^{13}$C-WSOCg. Organic nitrate and
organic sulfate generated by aged WSOCg may play an important role in gas-particle conversion
of nitrate and sulfate. However, it was hard to distinguish contribution by aging of WSOCp, gas
conversion of SVOC and liquid phase formation of WSOCp in gas-particle conversion of
ammonium, for both of them can lead to enrichment of $\delta^{13}$C-WSOCp. Increase of fp-$NO_2^-$ leaded
to enrichment of $\delta^{13}$C-WSOCp and dilution of $\delta^{13}$C-WSOCg on an extremely significant level
($p<0.01$). Particle conversion of SVOC, liquid phase generation of WSOCp and aging of WSOCp
may enhanced in rapid gas-particle conversion of $NO_2^-$. However, this was in contradiction with
$\delta^{13}$C-WSOCp depleted when concentration of $NO_2^-$ increased. $NO_2^-$ may have other relatively
important formation pathways apart from gas-particle conversion from HONO, such as
heterogeneous absorption of $NO_2$ by acetate in WSOCp to generate $NO_2^-$(Wen-Xiu, et al., 2022).
Increase of fp-acetate leaded to increase of WSOCp on an extremely significant level ($p<0.01$), with
none correlation with $\delta^{13}$C-WSOC. Acetate may be the main component of WSOCg which
converses to particle. Increase of fp-oxalate leaded to enrichment of $\delta^{13}$C-WSOCg on a significant
level ($p<0.05$), with none correlation with concentration of WSOC. Oxalate may be an important
component of WSOCg which is easier to aging.
3.5 Effects of meteorological conditions
Increase of wind speed promoted enrichment of $\delta^{13}$C-WSOCp and $\delta^{13}$C-WSOCg on an
extremely significant level ($p<0.01$), with no effects on isotope fractionation and concentration of
WSOC in two phases. However, change of wind direction could affect fp-WSOC and $\delta^{13}$C-WSOCg
on an extremely significant level ($p<0.01$). It was believed that there was a source around sampling
site with depleted $\delta^{13}$C. WSOC transmitted under high wind speed environment mixed with local
pollutants, leading to a different source composition. However, gas-particle distribution mechanism
of WSOC and emission intensity of pollution sources unchanged. For accidental effects of wind
direction was filtered by mean daily analysis, it's believed there existed a strong source of WSOC
in the northwest wind direction.
Among the other meteorological elements, the increase of temperature promoted enrichment
of $\delta^{13}$C-WSOCp at a significant level ($p<0.05$). Increase of atmospheric pressure, RH and radiation
intensity promoted the enrichment of $\delta^{13}$C-WSOCg above a significant level ($p<0.05$). The increase
of relative humidity, air pressure and radiation may promote the aging of WSOCg. Though
conversion of WSOCp to WSOCg can also lead to an enriched $\delta^{13}$C-WSOCg, a significant positive
correlation between RH and WSOCp ($p<0.05$) indicated a high possibility that intensity of specific
emission source changed in environment with high RH. There was none significant correlation
between temperature and concentration of WSOC, indicating that the change of temperature may
has no effect on emission intensity. Increase of temperature may be caused enrichment of $\delta^{13}$C-
WSOCp by promoting more conversion of SVOC to WSOCg and accelerating the aging of WSOCp

4.   Conclusion

In this study, we developed a high time resolution method for determining the $\delta^{13}$C values of

WSOCp and WSOCg by combination of wet oxidation pretreatment and IRMS. With improvement
of oxidation method and determination method, $\delta^{13}$C value of liquid sample with a carbon content
between 0.5 to 5μg can be determined with an accuracy of 0.6 ‰. Using this method, the $\delta^{13}$C value
of WSOCp and WSOCg in winter of 2021 at an urban site of Nanjing were determined, which were
-25.9±0.7 ‰ and -29.9±0.9 ‰ respectively. Approaching $\delta^{13}$C of WSOCp and WSOCg indicted a
common source during heavy haze period, which may be liquid fossil fuel.

The fractionation between $\delta^{13}$C-WSOCp and $\delta^{13}$C-WSOCg had a significant diurnal variation

of low in the day and high in the night, reaching the lowest at early noon. The existence of sunlight
determined whether the formation of WSOCp and WSOCg was an independent chemical process,
but there was no correlation between isotope signal and radiation intensity. The fractionation was
restricted by WSOCg, RH, $O_3$ and temperature during daytime, and was restricted by $O_3$, $PM_{2.5}$ and
$PM_{10}$ during nighttime. Rise of WSOCg and RH were negative factor for isotope fractionation, and
rise of temperature, $O_3$, $PM_{2.5}$ and $PM_{10}$ were positive factor for isotope fractionation.

Gas-particle distribution mechanism of WSOC was restricted by meteorological conditions,

precursor gaseous pollutants, gaseous oxidants and acid gases, which can lead to isotope dilution
and enrichment by affecting gas-particle distribution of SVOC, gas-liquid phase reaction proportion
of WSOCp, aging process of WSOCp and WSOCg and gas-particle exchange of WSOC. Gas-
particle distribution of SVOC was restricted by $NO_2$, $O_3$, gaseous oxalic acid and temperature; Gas-
liquid phase reaction proportion of WSOCp was restricted by HONO and $NO_2$; Aging process of





453 WSOCp was restricted by $O_3$ and temperature; Aging process of WSOCg was restricted by CO,

454 $NH_3$, HCl, RH, pressure and radiation; Gas-particle exchange of WSOC was restricted by $NH_3$, HCl

455 and HONO.

456  In the components of $PM_{2.5}$, generation of SNA had significant correlation with gas-particle

457 distribution of WSOC. The formation of SNA and WSOCp was accompanied by the aging of

458 WSOCg and the enhancement of liquid phase reaction of WSOCp. Generation of haze may be

459 related to retention and aging of WSOCg. Reaction between HONO and $NO_2^-$ may also have some

460 unknown effects on generation of WSOC through gas phase reaction.

461

462 **Author Contribution**

463 Hao-Ran Yu and Yan-Lin Zhang designed the experiments; Hao-Ran Yu adapted the instruments,

464 carried out isotope test; Fang Cao, Xiao-Ying Yang, Tian Xie and Yu-Xian Zhang organized the

465 sampling, Yongwen Xue carried out TOC test. Hao-Ran Yu prepared the manuscript with

466 contributions from all co-authors.

467

468 **Competing Interests**

469 The authors declare that they have no conflict of interest.

470

471 **Acknowledgements**

472  This work was supported by the National Natural Science Foundation of China (No. 41977305)

473 and Jiangsu Innovation & Entrepreneurship Team. The authors declare no competing financial

474 interest.

475

476 **Reference**

477 Aggarwal S. G., Kawamura K.: Molecular distributions and stable carbon isotopic compositions of
478 dicarboxylic acids and related compounds in aerosols from Sapporo, Japan: Implications for
479 photochemical aging during long-range atmospheric transport, Journal of Geophysical Research, 113:
480 D14301, 10.1029/2007jd009365, 2008.
481 Alexander Becky, Sherwen Tomas, Holmes Christopher, Fisher A. Jenny, Chen Qianjie, Evans Mat,
482 Kasibhatla Prasad: Global Inorganic Nitrate Production Mechanisms: Comparison of a Global Model



with Nitrate Isotope Observations, Atmospheric Chemistry and Physics, 20: 3859–3877, 2019.
Anderson Casey H., Dibb Jack E., Griffin Robert J., Hagler Gayle S. W., Bergin Michael H.: Atmospheric
water-soluble organic carbon measurements at Summit, Greenland, Atmospheric Environment, 42(22):
5612-5621, 10.1016/j.atmosenv.2008.03.006, 2008.
Anderson R. S., Iannone R., Thompson A. E., Rudolph J., Huang L.: Carbon kinetic isotope effects in
the gas-phase reactions of aromatic hydrocarbons with the OH radical at 296 +/- 4 K, , Geophysical
Research Letters, 31: 15108, 10.1029/2004gl020089, 2004.
August Andersson, Junjun Deng, Ke Du, Mei Zheng, Caiqing Yan, Martin SköLd, ÖRjan Gustafsson:
Regionally-Varying Combustion Sources of the January 2013 Severe Haze Events over Eastern China,
Environment Science & Technology, 49: 2038-2043, 10.1021/es503855e, 2015.
Bond T. C., Doherty S. J., Fahey D. W., Forster P. M., Berntsen T., Deangelo B. J., Flanner M. G., Ghan
S., Karcher B., Koch D., Kinne S., Kondo Y., Quinn P. K., Sarofim M. C., Schultz M. G., Schulz M.,
Venkataraman C., Zhang H., Zhang S., Bellouin N., Guttikunda S. K., Hopke P. K., Jacobson M. Z.,
Kaiser J. W., Klimont Z., Lohmann U., Schwarz J. P., Shindell D., Storelvmo T., Warren S. G., Zender
C. S.: Bounding the role of black carbon in the climate system: A scientific assessment, Journal of
Geophysical Research, 118: 5380-5552, 10.1002/jgrd.50171, 2013.
Cao J. J., Chow J. C., Tao J., Lee S. C., Watson J. G., Ho K. F., Wang G. H., Zhu C. S., Han Y. M. : Stable
carbon isotopes in aerosols from Chinese cities: Influence of fossil fuels Atmospheric Environment, 45:
1359-1363, 10.1016/j.atmosenv.2010.10.056, 2011.
Carlton A. G., Turpin B. J. : Particle partitioning potential of organic compounds is highest in the Eastern
US and driven by anthropogenic water, Atmospheric Chemistry and Physics, 13: 10203-10214,
10.5194/acp-13-10203-2013, 2013.
Carlton A. G., Wiedinmyer C., Kroll J. H.: A review of Secondary Organic Aerosol (SOA) formation
from isoprene, Atmospheric Chemistry and Physics, 9: 4987-5005, 10.5194/acp-9-4987-2009, 2009.
Chi Yang, Shuxin Zhou, Chunyan Zhang, Mingyuan Yu, Fang Cao, Yanlin Zhang: Atmospheric
Chemistry of Oxalate: Insight Into the Role of Relative Humidity and AerosolAcidity From High-
Resolution Observation, Journal of Geophysical Research: Atmospheres, 127(4): e2021JD035364,
10.1029/2021jd035364, 2022.
El-Sayed Marwa M. H., Wang Yingqing, Hennigan Christopher J.: Direct atmospheric evidence for the
irreversible formation of aqueous secondary organic aerosol, Geophysical Research Letters, 42(13):
5577-5586, 10.1002/2015gl064556, 2015.
Fisseha R., Saurer M., Jaggi M., Szidat S., Siegwolf R. T., Baltensperger U.: Determination of stable
carbon isotopes of organic acids and carbonaceous
aerosols in the atmosphere, Rapid Commun Mass Spectrom, 20(15): 2343－2347, 10.1002/rcm.2586,

517   2006.

Fu P., Kawamura K., Usukura K., Miura K.: Dicarboxylic acids, ketocarboxylic acids and glyoxal in the
marine aerosols collected during a round-the-world cruise, Marine Chemistry, 148: 22-32,
10.1016/j.marchem.2012.11.002, 2013.
Gehui Wang, Fang Zhang, Jianfei Peng, Lian Duan, Yuemeng Ji, Wilmarie Marrero-Ortiz, Jiayuan Wang,
Jianjun Li, Can Wu, Cong Cao, Yuan Wang, Jun Zheng , Jeremiah Secrest, Yixin Li, Yuying Wang, Hong
Li, Na Li, Renyi Zhang: Particle acidity and sulfate production during severe haze events in China cannot
be reliably inferred by assuming a mixture of inorganic salts, Atmospheric Chemistry and Physic, 18:
10123-10132, 10.5194/acp-18-10123-2018, 2018.
Gehui Wang, Renyi Zhang, Mario E. Gomez, Lingxiao Yang, Misti Levy Zamora, Min Hu, Yun Lin,


Jianfei Peng, Song Guo, Jingjing Meng, Jianjun Lia, Chunlei Cheng, Tafeng Hu, Yanqin Ren, Yuesi Wang,
Jian Gao, Junji Cao, Zhisheng An, Weijian Zhou, Guohui Lia, Jiayuan Wang, Pengfei Tian, Wilmarie
Marrero-Ortiz, Jeremiah Secrest, Zhuofei Du, Jing Zheng, Dongjie Shang, Limin Zeng, Min Shao,
Weigang Wang, Yao Huang, Yuan Wang, Yujiao Zhu, Yixin Li, Jiaxi Hu, Bowen Pan, Li Cai, Yuting
Cheng, Yuemeng Ji, Fang Zhang, Daniel Rosenfeld, Peter S. Liss, Robert A. Duce, Charles E. Kolb,
Mario J. Molina: Persistent sulfate formation from London Fog to Chinese haze, Proceedings of the
National Academy of Sciences of the United States of America, 113(48): 13630–13635,
10.1073/pnas.1616540113, 2016.
Gensch I., Kiendler-Scharr A., Rudolph J., .    206–221. Htps:/Doi.Org/: Isotope ratio studies of
atmospheric organic compounds: principles, methods, applications and potential, International Journal
of Mass Spectrometry, 365–366: 206-221, 10.1016/j.ijms.2014.02.004., 2014.
Hennigan C. J., Bergin M. H., Russell A. G., Nenes A., Weber R. J.: Gas/particle partitioning of water-
soluble organic aerosol in Atlanta, Atmospheric Chemistry and Physics, 9: 3613–3628, 10.5194/acp-9-
540    3613-2009, 2009.
Ho K. F., Cao J. J., Lee S. C., Kawamura K., Zhang R. J., Chow J. C., Watson J. G.: Dicarboxylic acids,
ketocarboxylic acids and dicarbonyls in urban atmosphere of China., Journal of Geophysical Research,
112: D22S27, 10.1029/2006JD008011, 2006.
Jingyue Tang, Liming Zeng, Huabin Dong: A new method for on-line measurements of water-solube
oranic compounds, Acta Scientiae Circumstantiae, 30(5): 908-914, 10.1631/jzus.A1000244, 2010.
Kondo Y., Miyazaki Y., Takegawa N., Miyakawa T., Weber R. J., Jimenez J. L., Zhang Q., Worsnop D.
R.: Oxygenated and water-soluble organic aerosols in Tokyo, Journal of Geophysical Research, 112:
D01203, 10.1029/2006jd007056, 2007.
Lewtas Joellen, Pang Yanbo, Booth Derrick, Reimer Steve, Eatough Delbert J., Gundel Lara A.:
Comparison of Sampling Methods for Semi-Volatile Organic Carbon Associated with PM2.5, Aerosol
Science and Technology, 34(1): 9-22, 10.1080/02786820118935, 2001.
Liu J. M., Zhang X. L., Parker E. T., Veres P. R., Roberts J. M., De Gouw J. A., Hayes P. L., Jimenez J.
L., Murphy J. G., Ellis R. A., Huey L. G., Weber R. J.: On the gas-particle partitioning of soluble organic
aerosol in two urban atmospheres with contrasting emissions: 2. Gas and particle phase formic acid,
Journal of Geophysical Research: Atmospheres, 117(D21): D00V21, 10.1029/2012JD017912, 2012.
Liu S., Ahlm L., Day D. A., Russell L. M., Zhao Y. L., Gentner D. R., Weber R. J., Goldstein A. H., Jaoui
M., Offenberg J. H., Kleindienst T. E., Rubitschun C., Surratt J. D., Sheesley R. J., Scheller S.: Secondary
organic aerosol formation from fossil fuel sources contribute majority of summertime organic mass at
Bakersfield,    Journal    of    Geophysical    Research:    Atmospheres,    117(D24):    D00V26,
10.1029/2012JD018170, 2012.
Lv S., Wang F., Wu C., Chen Y., Liu S., Zhang S., Li D., Du W., Zhang F., Wang H., Huang C., Fu Q.,
Duan Y., Wang G.: Gas-to-Aerosol Phase Partitioning of Atmospheric Water-Soluble Organic
Compounds at a Rural Site in China: An Enhancing Effect of NH3 on SOA Formation, Environmental
Science & Technology, 56(7): 3915-3924, 10.1021/acs.est.1c06855, 2022.
Meng J. J., Wang G. H., Li J. J., Cheng C. L., Ren Y. Q., Huang Y., Cheng Y. T., Cao J. J., Zhang T.:
Seasonal characteristics of oxalic acid and related SOA in the free troposphere of Mt. Hua, central China:
Implications for sources and formation mechanisms, Science of Total Environment, 493: 1088-1097,
10.1016/j.scitotenv.2014.04.086, 2014.
Miyazaki Y., Kondo Y., Takegawa N., Komazaki Y., Fukuda M., Kawamura K., Mochida M., Okuzawa
K., Weber R. J.: Time-resolved measurements of water-soluble organic carbon in Tokyo, Journal of





Geophysical Research, 111: D23206, 10.1029/2006JD007125, 2006.
Noziere B., Kalberer M., Claeys M., Allan J., D'anna B., Decesari S., Finessi E., Glasius M., Grgic I.,
Hamilton J. F., Hoffmann T., Iinuma Y., Jaoui M., Kahnt A., Kampf C. J., Kourtchev I., Maenhaut W.,
Marsden N., Saarikoski S., Schnelle-Kreis J., Surratt J. D., Szidat S., Szmigielski R., Wisthaler A.: The
molecular identification of organic compounds in the atmosphere: state of the art and challenges,
Chemical Reviews, 115: 3919-3983, 10.1021/cr5003485, 2015.
Pavuluri C. M., Kawamura K., Swaminathan T., Tachibana E.: Stable carbon isotopic compositions of
total carbon, dicarboxylic acids and glyoxylic acid in the tropical Indian aerosols: Implications for
sources and photochemical processing of organic aerosols, Journal of Geophysical Research, 116:
D18307, 10.1029/2011jd015617, 2011.
Pöschl U.: Atmospheric aerosols: composition, transformation, climate and health effects, Angewandte
Chemie-International Edition, 44: 7520-7540, 10.1002/anie.200501122, 2005.
Rudolph J., Anderson R. S., Czapiewski K. V., Czuba E., Ernst D., Gillespie T., Huang L., Rigby C.,
Thompson A. E.: The stable carbon isotope ratio of biogenic emissions of isoprene and the potential use
of stable isotope ratio measurements to study photochemical processing of isoprene in the atmosphere,
Journal of Atmospheric Chemistry, 44,: 39-55, 10.1023/A:1022116304550, 2003.
Rudolph J., Czuba E., Huang L.: The stable carbon isotope fractionation for reactions of selected
hydrocarbons with OH-radicals and its relevance for atmospheric chemistry, Journal of Geophysical
Research, 105(D24): 29329-29346, 10.1029/2000JD900447, 2000.
Saehee Lim, Xiaoyang Yang, Meehye Lee, Gang Li, Yuanguan Gao, Xiaona Shang, Kai Zhang, Claudia
I. Czimczik, Xiaomei Xu, Min-Suk Bae, Kwang-Joo Moon, Kwonho Jeon: Fossil-driven secondary
inorganic PM2.5 enhancement in the North China Plain: Evidence from carbon and nitrogen isotopes,
Environmental Pollution, 266: 115163, 10.1016/j.envpol.2020.115163, 2020.
Sareen N., Waxman E. M., Turpin B. J., Volkamer R., Carlton A. G.: Potential of Aerosol Liquid Water
to Facilitate Organic Aerosol Formation: Assessing Knowledge Gaps about Precursors and Partitioning,
Environmental Science & Technology, 51: 3327-3335, 10.1021/acs.est.6b04540, 2017.
Shaojun Lv, Fanglin Wang, Can Wu, Yubao Chen, Shijie Liu, Si Zhang, Dapeng Li, Wei Du, Fan Zhang,
Hongli Wang, Cheng Huang, Qingyan Fu, Yusen Duan, Gehui Wang: Gas-to-Aerosol Phase Partitioning
of Atmospheric Water-Soluble Organic Compounds at a Rural Site in China: An Enhancing Effect of
NH3 on SOA Formation, Environmental Science & Technology, 56(7): 3915-3924,
10.1021/acs.est.1c06855, 2022.
Suto N., Kawashima H.: Online wet oxidation/isotope ratio mass spectrometry method for determination
of stable carbon isotope ratios of water-soluble organic carbon in particulate matter, Rapid Commun
Mass Spectrom, 32(19): 1668-1674, 10.1002/rcm.8240, 2018.
Vodicka P., Kawamura K., Schwarz J., Zdimal V.: Seasonal changes in stable carbon isotopic composition
in the bulk aerosol and gas phases at a suburban site in Prague, Science of Total Environment, 803:
149767, 10.1016/j.scitotenv.2021.149767, 2022.
Wang G. H., Kawamura K., Cheng C. L., Li J. J., Cao J. J., Zhang R. J., Zhang T., Liu S. X., Zhao Z. Z.:
Molecular Distribution and Stable Carbon Isotopic Composition of Dicarboxylic Acids, Ketocarboxylic
Acids, and alpha-Dicarbonyls in Size-Resolved Atmospheric Particles From Xi'an City, China,
Environmental Science & Technology, 46: 4783-4791, 10.1021/es204322c, 2012.
Weber R. J., Sullivan A. P., Peltier R. E., Russell A., Yan B., Zheng M., Gouw J., Warneke C., Brock C.,
Holloway J. S., Atlas E. L., Edgerton E.: A study of secondary organic aerosol formation in the
anthropogenic-influenced southeastern United States, Journal of Geophysical Research, 112: D13302,



10.1029/2007JD008408, 2007.
Wen-Xiu Pei, Shuai-Shuai Ma, Zhe Chen, Yue Zhu, Shu-Feng Pang, Yun-Hong Zhang: Heterogeneous
uptake of NO2 by sodium acetate droplets and secondary nitrite aerosol formation, Journal of
Environmental Sciences, : 10.1016/j.jes.2022.05.04, 2022.
Yang Li, Shang Yue, Hannigan Michael P., Zhu Rui, Wang Qin'geng, Qin Chao, Xie Mingjie: Collocated
speciation of PM2.5 using tandem quartz filters in northern nanjing, China: Sampling artifacts and
measurement uncertainty, Atmospheric Environment, 246: 10.1016/j.atmosenv.2020.118066, 2021.
Zhang Wenqi, Zhang Yan-Lin, Cao Fang, Xiang Yankun, Zhang Yuanyuan, Bao Mengying, Liu Xiaoyan,
Lin Yu-Chi: High time-resolved measurement of stable carbon isotope composition in water-soluble
organic aerosols: method optimization and a case study during winter haze in eastern China, Atmospheric
Chemistry and Physics, 19(17): 11071-11087, 10.5194/acp-19-11071-2019, 2019.
Zhang X. L., Liu J. M., Parker E. T., Hayes P. L., Jimenez J. L., De Gouw J. A., Flynn J. H., Grossberg
N., Lefer B. L., Weber R. J.: On the gas-particle partitioning of soluble organic aerosol in two urban
atmospheres with contrasting emissions: 1. Bulk water-soluble organic carbon, Journal of Geophysical
Research: Atmospheres, 117(D21): D00V21, 10.1016/j.atmosenv.2013.04.060, 2012.
Zhang Y. L., Kawamura K., Cao F., Lee M.: Stable carbon isotopic compositions of low‐molecular‐
weight dicarboxylic acids, oxocarboxylic acids, α‐dicarbonyls, and fatty acids: Implications for
atmospheric processing of organic aerosols, Journal of Geophysical Research, 121: 3707-3717,
10.1002/2015jd024081, 2016.