# Peer review of "High-resolution observation of stable carbon isotope ratios of water-soluble organic"

_Atmospheric Measurement Techniques, 2022_

## Author Comment (AC1)

**Response to reviewers' comments**

**(Manuscript No. amt-2022-239 RC1)**

**Anonymous Referee #1:**

This study developed a new method and system to determine stable carbon isotope in both the water-soluble organic carbon in the gaseous and particle phases. The novelty of this study lies in the method and its potential application. They also found the difference in WSOC mass concentration and the isotope composition between the day and night samples. Although the explanation of the influencing factors for the differences remains unresolved, I strongly recommend it for a quick publication after they can address the following specific comments.

Dear reviewer:

Thank you very much for your comments and advises on our paper! They have been encouraging and constructive. We have learnt quite a lot from it.

After carefully studying the comments and advises, we have made corresponding changes to our paper.

*1.Line 26-27: the sentence was not clear.*

*The authors should use the isotope expression more carefully. For examples: lower, higher, depleted or enriched. For example, to say that "one sample is enriched in 15N relative to another because of ..." is proper usage. Phrases such as "a sample has an enriched d15N value" are misuses of terminology.*

**Author's response:**

Thanks for the reviewer's comments. Sentence in line 26-27 is modified to "It was found that WSOCg has significant higher concentration than WSOCp, and is depleted in $\delta^{13}C$ relative to WSOCp.".

*2.Line 38-40: rewording is necessary*

**Author's response:**

Thanks for the reviewer's comments. Description in line 38-40 is modified to "Generations of WSOCp and WSOCg during nighttime are two independent processes, and the presence of radiation decides conversion between WSOCp and WSOCg, for $\delta^{13}C$-WSOC of two phases showed significant correlation only during daytime.".

*3.Line 149: details are needed here,*

*Was the method for 13C in WSOC improved or optimized compared to the previous method. If so, the details should be given or highlighted.*

**Author's response:**

Thanks for the reviewer's comments. Details in line 149 is suppled as "WSOCp was sampled by filters at the same time to compare the sampling efficiency of IGAC system. The daily average of WSOCp sampled by filters was 24.1μg/m³, while WSOCp determined by the IGAC system sampled at the same time was 29.7μg/m³.". And the detailed data is added to Supplement

Information Table S9.

Table S9 Comparison of WSOCp sampled by filter and by IGAC system

| Time | IGAC WSOCp (μg/m3) | Filter WSOCp (μg/m3) |
|---|---|---|
| 2021/12/10 1:00 | 31.75 | 15.99 |
| 2021/12/11 2:00 | 26.57 | 17.45 |
| 2021/12/12 3:00 | 21.23 | 19.04 |
| 2021/12/13 4:00 | 20.84 | 22.32 |
| 2021/12/14 5:00 | 26.96 | 19.38 |
| 2021/12/15 6:00 | 29.40 | 20.35 |
| 2021/12/16 7:00 | 28.34 | 21.32 |
| 2021/12/17 8:00 | 27.17 | 19.14 |
| 2021/12/18 9:00 | 25.65 | 21.89 |
| 2021/12/19 10:00 | 33.37 | 21.30 |
| 2021/12/20 11:00 | 32.63 | 16.38 |
| 2021/12/21 12:00 | 32.25 | 27.41 |
| 2021/12/22 13:00 | 31.11 | 47.91 |
| 2021/12/23 14:00 | 31.49 | 35.49 |
| 2021/12/24 15:00 | 31.87 | 23.08 |
| 2021/12/25 16:00 | 26.76 | 18.91 |
| 2021/12/26 17:00 | 30.68 | 29.84 |
| 2021/12/27 18:00 | 31.76 | 24.55 |
| 2021/12/28 19:00 | 34.64 | 20.36 |
| 2021/12/29 20:00 | 37.51 | 22.18 |
| 2021/12/30 21:00 | 39.22 | 37.75 |

For the details of this method. It's an improved method for $\delta^{13}C$ in WSOC compared to the previous method both in instrument transformation and correction method improvement. Details are given in chapter 3.1 Improvement of determination method, as "Compared with previous determination method (Zhang, et al., 2019), the determination limit was improved to less than 5μgC in this study. The headspace bottle is pre purged before sample injection and liquid content of sample injection is unified as 2mL to reduce the influence of carbon dioxide dissolved in water. The composition of oxidizer is changed to reduce the influence of carbon blank in oxidizer."

*4.The influences of the metrological condition and chemical composition should be reordered. the mechanism for the WSOCg-WSOCp distribution was not well explained and further discussion is needed.*

**Author's response:**

Thanks for the reviewer's comments. The influences of the meteorological condition and chemical composition is reordered. Discussion now is reordered as 3.3 Stable carbon isotopic variation characteristic, 3.4 The influence of meteorological condition, 3.5 The influence of gaseous pollutants, and 3.6 The indication by chemical composition of particle matters. The influence of radiation which used to be discussed in chapter 3.3 Stable carbon isotopic variation characteristic is moved to chapter 3.4 The influence of meteorological condition.

The mechanism for the WSOCg-WSOCp distribution is rediscussed in chapter 3.3, chapter 3.4, chapter 3.4, chapter 3.5, and chapter 3.6.

In chapter 3.3, five processes of WSOCg-WSOCp distribution generating isotopic fraction of $\delta^{13}C$ between the gas phase and particle phase of WSOC were discussed, including "isotope fractionation due to the gas-particle distribution of SVOC (Vodicka, et al., 2022); isotope fractionation due to equilibrium exchange of WSOCp and WSOCg (Gensch, et al., 2014); isotope changes caused by source composition changes and emission intensity changes of WSOCp and WSOCg (Saehee, et al., 2020); enrichment of $\delta13C$ caused by retention and aging of WSOCp and WSOCg; isotope fractionation due to proportion change of gas phase reaction and liquid phase reaction in the generation process of WSOCp."

In chapter 3.4, influences of wind speed, radiation intensity, temperature and RH were rediscussed separately. It is concluded as "The liquid phase reaction of WSOCg to WSOCp may be dominant in the daytime, and the oxidative aging of WSOCp may be dominant at night. The existence of sunlight rather than its intensity determined whether the formation of WSOCp and WSOCg were independent chemical processes. Temperature and radiation may accelerate the aging process of WSOCp."

In chapter 3.5, influences of CO, NH$_3$, O$_3$, NO$_2$ and gaseous acids were rediscussed. It is concluded that CO has influence on sources of WSOC while no influence on WSOCg-WSOCp distribution. NH$_3$ has both influence on sources of WSOC and WSOCg-WSOCp distribution. O$_3$, NO$_2$ and gaseous acids has influence on WSOCg-WSOCp distribution while no influence on sources of WSOC. It is concluded as "O$_3$ and NO$_2$ promote the conversion to particle phase and the aging process of WSOCp, NH$_3$ promotes the retention and aging process of gas phase."

In chapter 3.6, we rediscussed the generation processes of different components in particle matters, and the relationship between them and WSOCg-WSOCp distribution processes. It is concluded as "the emission intensity of WSOC sources increased while the source composition remains stable under a high concentration of particle matters." and "conversion of WSOCg to WSOCp may be along with SNA generation process."

Based on the discussions, conclusion is rewrite as "In this study, we developed an improved high-time resolution method for determining the $\delta^{13}C$ values of WSOCp and WSOCg by a combination of wet oxidation pretreatment and IRMS. With the improvement of the oxidation method and determination method, the $\delta^{13}C$ value of the liquid sample with a carbon content between 0.5 to 5μg can be determined with an accuracy of 0.6 ‰. Using this method, the $\delta^{13}C$ value of WSOCp and WSOCg in the winter of 2021 at an urban site in Nanjing were determined, which were -25.9 ± 0.7 ‰ and -29.9 ± 0.9 ‰ respectively. Approaching $\delta^{13}C$ of WSOCp and WSOCg indicated a common source during the heavy haze period, which may be liquid fossil fuel.

The fractionation between $\delta^{13}C$-WSOCp and $\delta^{13}C$-WSOCg had a significant diurnal variation of low in the day and high in the night, reaching the lowest at early noon. The change of fractionation mainly come from gas-particle distribution of SVOC, equilibrium exchange of WSOCp and WSOCg, source composition and emission intensity changes of WSOCp and WSOCg, retention and aging of WSOCp and WSOCg, proportion change of gas phase reaction and liquid phase reaction in the generation process of WSOCp.

The liquid phase reaction of WSOCg to WSOCp may be dominant in the daytime, and the oxidative aging of WSOCp may be dominant at night. The existence of sunlight rather than its intensity determined whether the formation of WSOCp and WSOCg were independent chemical

processes. Temperature and radiation may accelerate the aging process of WSOCp. $O_3$ and $NO_2$ promote the conversion to particle phase and the aging process of WSOCp, $NH_3$ promotes the retention and aging process of gas phase. Indicated by response of chemical composition in particle matters, conversion of WSOCg to WSOCp may be along with SNA generation process."

Thank you very much for the excellent and professional revision of our paper!

---

## Author Comment (AC3)

**Response to reviewers' comments**

**(Manuscript No. amt-2022-239 RC3)**

**Anonymous Referee #2:**

This paper presents ambient, simultaneous WSOC concentration measurements in gas and particle phase, sampled in summer and winter in Nanjing, China. For winter, isotopic ratios for bulk WSOC in both phases are given. It aims to interpret concentration and d13C differences for the two phases. The use of carbon isotope ratios for the study of atmospheric pollution and the chemistry of organic compounds in the atmosphere is a newly emerging tool. Yet, there is hitherto relatively little information on isotopic signatures of sources and less understanding for the processes altering the pollutants from emission to sampling. Therefore, such data, as showed here, are valuable. Nevertheless, they have to be presented and interpreted in a proper way prior publishing.

Dear reviewer:

Thank you very much for your comments and advises on our paper! Those comments are valuable and very helpful. We have read through comments carefully and have made corrections.

**General comments**

*1) One main criticism to this manuscript is related to the methodological part.*

*Section 3.1 promises to describe the 'Improvement of determination method'. What was concretely improved compared to the existing methods?*

*Further, Fig 5b is redundant. It is showing that the slope of the line describing d13C measured by the method vs standard d13C is increasing when the blank becomes more important, which should be already clear from the mathematical point of view. Then, do the fitted lines to the measurements (described by logarithmic relations?) make any physical sense? Remove, give instead the relation for the sample d13C dependence on the blank (Eq. 1 in Fisseha et al 2006, or Eq1-7 in Zhang et al. 2019). Anyhow, one should take any of these two studies as an example for describing a method development (including tests of the standard samples recovery, quality control and quality assurance procedures...)*

**Author's response:**

Thanks for the reviewer's comments. Methodological part is recognized as '2.2.1 Sample pretreatment and isotope determination', '2.2.2 Calibration of isotope results', '2.2.3 Quality control and quality assurance', '2.2.4 Improvement of determination method'. And tests of the standard sample recovery is applied in section 2.2.3.

In section 2.2.4, improvement of determination method is given in two aspects, procedural processing and equipment transformation, as 'Determined by previous method (Zhang, et al., 2019), $CO_2$ blank is 0.8μgC (2mL sample volume). $CO_2$ blank in the headspace bottle mainly came from three parts. Apart from OC of oxidizer/acidifier (Fisseha, et al., 2006), $CO_2$ dissolved in Milli-Q water and residual air $CO_2$ during purging result in the other parts of $CO_2$ blank. During the $CO_2$ blank test, the signal response of the mass spectrometer to samples was 80 mVs/μgC. Compared with helium purging under liquid level after sample injection, a method of helium purging before sample injection could effectively eliminate the impact of residual $CO_2$ in the air (< 2 mVs). In addition, the $CO_2$ blank signal would increase around 10 mVs (about 0.1μgC) with each 50mg

increase of $K_2S_2O_8$. And $CO_2$ blank signal would increase around 20 mVs (about 0.25μgC) with each 1mL increase of liquid in the headspace bottle. In contrast, $CO_2$ blank is more affected by liquid content. And this part of the $CO_2$ blank couldn't be controlled by the He purging or pre-heating. By controlling the sample volume, and reducing the oxidizer concentration and injection volume, the total $CO_2$ blank signal finally reached around 30 mVs (about 0.3μgC), approximately 19% of the average carbon content of the WSOC sample.

The Gasbench-IRMS system and determination method were improved in this method. The system used high-purity helium as carrier gas. Sample gas was pushed through the water trap (magnesium perchlorate) and VOC trap in the preconcentration unit (Precon) by helium at a pressure of 1.7 bar. After 260s of freeze enrichment and impurity removal in a liquid nitrogen trap, helium at a pressure of 0.6 bar was switched by rotating the six-way valve in Precon, pushing sample gas into the gas chromatographic column (Polra PLOT Q) to separate $N_2O$ and $CO_2$. The back purge valve of the front pipeline was opened at the same time to purge the sample injection pipeline. Finally, the sample gas entered IRMS for $\delta^{13}C$ determination after water removal through a Nafion permeation tube. It took a total of 24 min in this method, and the determination accuracy can reach 0.3 ‰ above 1μgC. Determined by previous method (Zhang, et al., 2019), the signal response of the mass spectrometer to samples was controlled to be 0.6 mVs/μgC. An average peak area of standard samples with 4μg C is improved from 3 mVs to 101 mVs in this method (n=3).'.

In section 2.2.2 Calibration of isotope results, combined with Fig.3 and Eq1-2, a calibration method different from Fisseha et al 2006 or Zhang et al. 2019 is introduced, as 'Determination values have obvious peak area dependence in a carbon content less than 5μgC (Fig. 3a), which is proved to be cause by the procedural blank contribution (Fisseha et al 2006, Zhang et al. 2019). Considering the solubility of different kinds of standard in water, three working standards were used in this study to establish the standard curve between the true value and determination value (Fig. 3b): potassium hydrogen phthalate (KHP) and two kinds of sucrose (Suc-1 and Suc-2). The carbon isotope composition of these three standards is analyzed by combustion method, using an elemental analyzer combined with an isotope ratio mass spectrometer (EA-IRMS, Thermo Fisher Scientific, USA), as follows: -12.08‰ (Suc-1), -24.83‰ (Suc-2), and -30.62‰ (KHP) (n=6). This range of $\delta^{13}C$ values can cover the majority of the $\delta^{13}C$-WSOC values in ambient air samples. Standards were resolved in Milli-Q water (resistivity 18.2MΩ) to make standard solutions of the carbon content of 0.5, 1, 2, and 4 μg in 2mL standard solution to test the procedures during the pretreatment. What's more, it was found in this study that over-heated oxidizer would cause severe depletion of $^{13}C$ in standard samples.

[Figure]

Fig. 3 Determination value of three isotopic working standards with different carbon contents. (a) Peak area dependent effect of $\delta^{13}C$; (b) Standard curve with different carbon content; (c) Slope correction curve of the standard curve; (d) Intercept correction curve of the standard curve.

Contributed by the procedural blank and isotope fractionation during the preparation, difference exists between determination value and actual value of the standard samples. The correction relation between the determination value and actual value can be expressed as linear equation as follow:

$$\delta^{13}C_{act}=k\times\delta^{13}C_{det}+b \qquad (Eq.1)$$

Where $\delta^{13}C_{act}$ is the isotope composition after the isotope calibration, $\delta^{13}C_{det}$ is the isotope composition determined by IRMS, k and b are the slope and the intercept obtained from the calibration curve.

As shown in Fig. 3b, correction relation changes with carbon content, which may be due to changes of relative contribution of the procedural blank in samples with different carbon content. As both the slope and the intercept have strong correlation with sample carbon content ($R^2$ =0.996), the correction relation between the determination value and actual value can be expressed as linear equation as follow:

$$\delta^{13}C_{act}=(0.2106\times\ln(C_{con})+0.5370)\times\delta^{13}C_{det}+(-4.8160\times\ln(C_{con})+6.9890) \qquad (Eq.2)$$

Where $C_{con}$ is the carbon content of samples.'

*2) given that the paper announces method improvement to measure very low amounts of WSOC, I miss a discussion on systematic errors. Please elaborate and based on that, what is the statistical significance of the day/night isotopic variations (are they still 'unimodal', 'bimodal')?*
**Author's response:**

Thanks for the reviewer's comments. Systematic errors are now elaborated in section 2.2.3, and discussion based on that is supplied as 'As discussed above, systematic error of $\delta^{13}C$-WSOC

caused by determination is less than 0.3‰. Random errors of $\delta^{13}$C-WSOCp and $\delta^{13}$C-WSOCg caused by sampling condition during the study period are 0.6±0.4‰ and 0.6±0.3‰, separately (Fig.4c). Compared with the variation range of $\delta^{13}$C-WSOCp and $\delta^{13}$C-WSOCg (Fig.4c, 2.3‰ and 2.0‰), systematic error only accounts for around 14%, and random errors account for an average of 29%. Based this, hourly average processing can filter random errors, clarifying diurnal changes of $\delta^{13}$C-WSOCp and $\delta^{13}$C-WSOCg. Besides, systematic error makes no significant influence in diurnal changes analysis.'

*3) Another concern is the way how the interpretation of these valuable ambient data is introduced. See for instance lines352-359:*

*'Increase of O3 leaded to enrichment of d13C-WSOCp and dilution of d13C-WSOCg on an extremely significant level (p<0.01). Similar with NO2, neither WSOCp nor WSOCg had correlation with concentration of O3, which means none variation of emission intensity of WSOC source when d13C changed. It was believed that SVOC conversed more to WSOCp under high concentration of O3, leading to of dilution of d13C-WSOCg. The aging of WSOCp was promoted at the same time, leading to enrichment of d13C-WSOCp. Dilution of d13C-WSOCg caused by conversion from SVOC to WSOCp exceeded enrichment of d13C-WSOCg caused by aging of WSOCg, leading to a more dilute d13C-WSOCg, indicating that O3 tends to react with SVOC rather than WSOCg'*

*Disregarding the extremely non-scientific used language, which makes very difficult for the reader to follow the text, the authors postulate in the beginning of this paragraph (and similarly throughout the whole discussion section) a causality which might be true or not. They combine, in this case, O3 with WSOC data. They observe a trend, here, increasing of O3 and WSOCp d13C and decreasing of WSOCg d13C. Their interpretation is: the increase in ozone leads to an increase in d13C in particles and decrease in gas-phase. Not always, when other processes causing opposite trends would act stronger. In the next lines, different hypotheses are eventually given, potential chemical and physical WSOC processes are described together with their impact on the isotopic ratios, which is mostly well done.*

*My recommendation: reorganize completely the discussion part 3.3 (lines259-430) and partially the conclusions part 4 (lines444-455) as following: (i) first describe the data trends, avoid at this point any conclusions. (ii)Further discuss the prevailing processes, mention potential changes in d13C due to these processes. (iii) Finally conclude using 'this might explain the observed trend'.*

**Author's response:**

Thanks for the reviewer's comments. the discussion part 3.3 (lines259-430) and partially the conclusions part 4 (lines444-455) is reorganized. Discussion now is reordered as '3.3 Stable carbon isotopic variation characteristic' and '3.4 Possible driving factors of stable carbon isotopic variation'. In section 3.3, stable carbon isotopic data trends are given. In section 3.4 five prevailing processes generating isotopic fraction of $\delta^{13}$C between the gas phase and particle phase of WSOC were concluded firstly, including 'isotope fractionation due to the gas-particle distribution of SVOC (Vodicka, et al., 2022); isotope fractionation due to equilibrium exchange of WSOCp and WSOCg (Gensch, et al., 2014); isotope changes caused by source composition changes and emission intensity changes of WSOCp and WSOCg (Saehee, et al., 2020); enrichment of $\delta$13C caused by retention and aging of WSOCp and WSOCg; isotope fractionation due to proportion change of gas phase reaction and liquid phase reaction in the generation process of WSOCp.'. Brief analysis results without any conclusions are given then, as 'As to $\delta^{13}$C fractionation between WSOCp and WSOCg

during the study period, there are significant negative correlations between it and CO, $NH_3$, WSOCg, $PM_{2.5}$, $PM_{10}$, relative humidity, along with significant positive correlations between it and $O_3$. As to $\delta^{13}C$ in WSOCp during the study period, there are significant negative correlations between it and $NO_2$, $NH_3$, WSOCg, nitrite, along with significant positive correlations between it and $O_3$, temperature, wind speed, chloride, sulfate, sodium. As to $\delta^{13}C$ in WSOCg during the study period, there are significant negative correlations between it and $O_3$, $NO_2$, along with significant positive correlations between it and CO, $NH_3$, WSOCp, WSOCg, $PM_{2.5}$, $PM_{10}$, radiation, relative humidity, air pressure, wind speed, fluorine, chloride, nitrate, sulfate, sodium, ammonium.'. All the factors are concluded as 'The meteorological condition affects partitioning between gas and particle phase by controlling the reaction condition. Along with gaseous precursor of WSOC, the gaseous pollutants participate gas phase reaction of WSOC directly. As important components of the particle matter apart from carbonaceous component, the ion composition can indicate the formation mechanism of WSOCp to a certain extent. Therefore, the following discussion is divided into three parts.' Further, section 3.4 is divided into '3.4.1 The influence of meteorological condition', '3.4.2 The influence of gaseous pollutants', and '3.4.3 The indication by chemical composition of particle matters', mentioning and discussing potential changes in $\delta^{13}C$ due to these processes. At last, it is concluded now as 'The liquid phase reaction of WSOCg to WSOCp may be dominant in the daytime, and the oxidative aging of WSOCp may be dominant at night. The existence of sunlight rather than its intensity determined whether the formation of WSOCp and WSOCg were independent chemical processes. Temperature and radiation may accelerate the aging process of WSOCp. $O_3$ and $NO_2$ may promote the condensation to particle phase and the aging process of WSOC, $NH_3$ may promotes the retention and aging process of gas phase. Indicated by response of chemical composition in particle matters, conversion of WSOCg to WSOCp may be along with SNA generation process.'

*4) To ease the overall understanding:*
*- a discussion in the introduction is mandatory, emphasizing the potential but also the limitations when using isotopic information. based on literature, mention all atmospheric processes impacting the WSOC isotopic ratios and indicate the linked variations in d13C depending on the ambient conditions. The studies of Kawamura's group are very useful for that goal.*
*Generally, reorganize the whole introduction. It is understandable that it contains a lot of information, but make it more systematic (separate the sources, mixing from physical and chemical processing)*
*- throughout the paper: descriptions such as 'conversion', 'transformation'.... should be replaced by specific terms used in the atmospheric research: volatilization, condensation, partitioning between gas- and particle phase...*

**Author's response:**

Thanks for the reviewer's comments.

-Discussion about $\delta^{13}C$ in introduction is reorganized into seven parts now, '$\delta^{13}C$-WSOC is influenced by source composition', '$\delta^{13}C$-WSOC is influenced by isotope fractionation', 'reported $\delta^{13}C$ fractionation in gas phase substances', 'reported $\delta^{13}C$ fractionation in particle phase substances', '$\delta^{13}C$ fractionation between the particle and gas phases differ at the level of individual carbon compounds', '$\delta^{13}C$ fractionation is influenced by meteorological conditions' and 'general conclusion of similar studies so far and significance of our study'. As is shown follow:

'The stable carbon isotope ratio ($^{13}C/^{12}C$, $\delta^{13}C$) can provide important information about

carbonaceous aerosol sources. $\delta^{13}C$ was generally used to distinguish sources of carbonaceous aerosols, such as biomass burning from C3 and C4 plants (Martinelli et al., 2002; Moura et al., 2008, Andersson et al., 2015), coal combustion (Gleason and Kyser, 1984; Widory, 2006, Andersson et al., 2015), vehicle exhaust (Widory, 2006) and liquid fossil sources (Andersson et al., 2015), for each source has different range of $\delta^{13}C$ value.

However, $\delta^{13}C$ has a mass-dependent isotope fractionation phenomenon, which is affected both by chemical reaction processes and physical processes, leading to an isotope variation apart from the influence of the source composition. Different processes usually lead to specific fractionation phenomenon of $\delta^{13}C$. Generally divided into equilibrium fractionation processes and kinetic fractionation processes, the kinetic isotope effect (KIE) occurs mainly during unidirectional reactions of organic substances, usually leading to an initial increase in $\delta^{13}C$ of precursors and a decrease in products (Vodicka et al, 2022). Equilibrium separation between phases determined by chemical processes will lead to the products be more enriched in $^{13}C$ in the particle phase than in the gas phase, while physical partitioning between the phases will make the difference be small (Gensch et al., 2014).

As to gas phase substances, the main scavenging pathway of VOCs is the reaction with OH radical and ozone. These atmospheric oxidants tend to react with VOCs with depleted $^{13}C$, resulting in the $^{13}C$ enrichment of residual VOCs in the atmosphere and $^{13}C$ depletion of oxidation products (Anderson et al., 2004; Rudolph et al., 2003; Rudolph et al., 2000). Besides, other reactions may lead to $^{13}C$ depletion in residual VOCs. For example, isotope fractionation in the process of biosynthesis of isoprene will lead to $\delta^{13}C$ of isoprene 2.6±0.9 ‰ smaller than it is in the blade (Rudolph et al., 2003).

As the secondary reaction of VOCs to generate SOA is an important source of WSOCp (Kondo et al., 2007; Weber et al., 2007), WSOCp usually have a more depleted $^{13}C$ than the precursor (Anderson et al., 2004; Rudolph et al., 2003; Rudolph et al., 2000). In the study of secondary particulate organic matter formed by the OH-radical induced photochemical oxidation of toluene in the gas phase, it is found that particulate organic matter is between 5.5 ‰ and 6.2 ‰ lighter than the precursor compound (Irei et al., 2006), and differs from primary particulate organic matter that often comes from petroleum related emissions (Irei et al., 2011). On the contrary, some studies have also shown that $^{13}C$ will be enriched during the aging process, such as a process in which binary acid reacts with OH and is removed in the form of $CO_2/CO$ (Aggarwal and Kawamura, 2008; Noziere et al., 2015; Pavuluri et al., 2011; Wang et al., 2012; Zhang et al., 2016). Apart from oxidation to highly oxygenated compounds, photochemical production of binary acid leads to $^{13}C$ enrichment as well (Pavuluri and Kawamura, 2016), while primarily emitted organic acids are less depleted in $^{13}C$ due to the absence of significant fractionation and therefore have a similar isotope ratio as their precursors (Sakugawa and Kaplan, 1995).

The fractionation of $\delta^{13}C$-WSOC between the particle and gas phases is difficult to be quantitative analyzed, for it may differ at the level of individual carbon compounds. Irei et al. (2006) reported that the particle-phase products of OH radical-induced reactions of toluene are 0.6 ± 0.2‰ lighter than the gas-phase products. Fisseha et al. (2009b) reported that the nopinone, an SOA formed by ozonolysis of β-pinene, is lighter by 2.3‰ in the gas phase than in aerosols. Saccon et al. (2015) reported that fractionation of $\delta^{13}C$ between particle phase and gas phase to be both negative and positive based on the kind of nitrophenol. Meusinger et al. (2017) reported a $\delta^{13}C$ fractionation range between −6.9 and +10.5‰ of α-pinene in SOAs.

Meteorological conditions affect the fractionation of $\delta^{13}$C-WSOC as well. Evidence has shown a temperature dependence of the KIE of individual compounds (Fisseha et al., 2009b; Gensch et al., 2011; Piansawan et al., 2017). Some studies have observed a significant negative correlation between $\delta^{13}$C-WSOC and temperature related to seasonal changes (Miyazaki et al., 2012; Vodicka et al., 2022). Besides, a stagnant atmosphere with low wind speed can creates favorable conditions for equilibrium fractionation of $^{13}$C between accumulated particles and the gas phase, which results in a larger difference in $\delta^{13}$C between these phases (Vodicka et al., 2022).

Seldom studies have reported the $\delta^{13}$C of gas phase carbonaceous substances in the atmosphere and partitioning between gas and particle phase based on $\delta^{13}$C. Meusinger et al. (2017) reported the $\delta^{13}$C of SOAs arising from α-pinene experiments in the particle and gas phases, and Vodicka et al. (2022) reported the $\delta^{13}$C of TC in particle and gas phases in the atmosphere in Central Europe. Measurement based on high-time resolution observation of $\delta^{13}$C-WSOCp and $\delta^{13}$C-WSOCg can serve to improve our understanding of the fundamental processes taking place between the particle and gas phases of WSOC. '.

-Descriptions of 'conversion' and 'transformation' are replaced throughout the paper now.

*5) the graphical abstract gives the impression that the 5.9 per mil difference between the isotopic ratios of gas and particle phase is due solely to 'circulation' (is here 'atmospheric transport' meant?). The authors should change the picture to make clear that ALL processes mentioned in lines 298-310 contribute (even at different degrees) to the observed fractionation.*

**Author's response:**

Thanks for the reviewer's comments. The graphical abstract is modified as follow. The observed fractionation is framed to avoid the impression that fractionation is due solely to 'circulation'. Besides, observed fractionation during the daytime and the nighttime is divided, corresponding to the conclusion that presence of radiation makes difference.

[Figure]

*6) throughout the manuscript and in the graphical abstract: d13C is a number which can only be big or small. Depletion (not 'dilution') and enrichment are used together with an isotope (12C or*

13C) to describe a change, e.g. in the course of a chemical reaction, the reactant becomes more and more enriched in 13C. Revise that everywhere!

**Author's response:**

Thanks for the reviewer's comments. Descriptions are revised to 'depletion of $^{13}$C in WSOC', 'enrichment of $^{13}$C in WSOC', 'depleted in $^{13}$C', 'enriched in $^{13}$C' in the graphical abstract and throughout the manuscript. Descriptions of 'dilution' are revised to 'depletion'.

*Specific comments:*

*1) The literature research for this manuscript is not satisfactory, should be thoroughly redone.*

*- lines107-111*

*'The main scavenging pathway of VOCs is its reaction with OH radical and ozone, and these atmospheric oxidants tend to react with VOCs depleted d13C (reverse dynamic isotope effect), resulting in the d13C enrichment of residual VOCs in the atmosphere and d13C dilution of particles as oxidation products (Anderson, et al., 2004; Rudolph, et al., 2003; Rudolph, et al., 2000).' The mentioned citations deal exclusively with gas-phase reactions! For compound specific isotope ratios in particles, check for instance the publications by Irei et al. What is the 'reverse dynamic isotope effect'? Remove that idiom throughout the manuscript.*

*- cite preferably the original publications*

*- line 301: replace Gensch et al. 2014 by Fisseha et al. 2009*

*- you might still use Gensch et al. in the introduction, since it gives valuable information on principles, potential and limitation of the isotopic research.*

*- remove Cao et al. throughout the manuscript and look for the original information (in Fig 4 by Cao, isotopic ratio ranges for C4 and C3 plants are presented the other way round. In the lines above, while the study by Martinelli et al. is suitable to cite, Moura et al. investigates the plant material isotopic ratios in sediments. There are a lot more appropriate studies on the isotopic ratio of plant material and emission related to that.)*

**Author's response:**

Thanks for the reviewer's comments.

-'reverse dynamic isotope effect' is removed throughout the manuscript. Studies deal with particle-phase reactions are supplied as 'In the study of secondary particulate organic matter formed by the OH-radical induced photochemical oxidation of toluene in the gas phase, it is found that particulate organic matter is between 5.5 ‰ and 6.2 ‰ lighter than the precursor compound (Irei et al., 2006), and differs from primary particulate organic matter that often comes from petroleum related emissions (Irei et al., 2011).'

- Gensch et al. 2014 is replaced by Fisseha et al. 2009 in line 301.

- Cao et al. is removed, and lines 99-101 is revised to '$\delta^{13}$C was generally used to distinguish sources of carbonaceous aerosols, such as biomass burning from C3 and C4 plants (Martinelli et al., 2002; Moura et al., 2008, Andersson et al., 2015), coal combustion (Gleason and Kyser, 1984; Widory, 2006, Andersson et al., 2015), vehicle exhaust (Widory, 2006) and liquid fossil sources (Andersson et al., 2015).'

*2) Linear regression analyses can be done when a linear relationship between two variables is expected (based on physics laws)*

*- lines275-286 and figure 5. Due to the complexity of the prevailing processes. in none of these cases*

*a linear regression analysis makes sense, neither the 'derived r2'.*

**Author's response:**

  Thanks for the reviewer's comments. Linear regression analyses and figure 5 are removed. Lines 275-286 is revised to 'Though there was no correlation between $\delta^{13}$C-WSOC fractionation and radiation, there was a negative correlation on an extremely significant level (p<0.01) between $\delta^{13}$C-WSOCp and $\delta^{13}$C-WSOCg in an environment with non-zero radiation. It indicated a highly relevant between generations of WSOCp and WSOCg during the daytime. Besides, the generations of WSOCp and WSOCg may be two independent processes during the nighttime.'.

*3) remove equation F1 (line 322)*

*This makes sense only in a compound specific study, where single SVOCs are measured.*

**Author's response:**

  Thanks for the reviewer's comments. Equation F1 (line 322) is removed.

==Other comments:==

*- please elaborate the source for the numbers in Table 1*

**Author's response:**

  Thanks for the reviewer's comments. The source for the numbers in Table 1 is elaborated as 'WSOC data comes from IGAC system samples determined by TOC-L; WSII data and organic acids data come from IGAC system samples determined by IC-5000; PM$_{2.5}$ data comes from online data of Pukou environmental supervising station.'

*- line205: 'However, determination values have obvious peak area independence in this range of carbon content (Fig. 3a).' On the contrary, Fig 3a shows that there is a d13C dependence, which was already shown 2006 by Fisseha et al. Revise!*

**Author's response:**

  Thanks for the reviewer's comments. The wrong description is revise as 'However, determination values still have obvious peak area dependence in a carbon content less than 5μgC (Fig. 3a).'.

Thank you very much for the kind work and professional advises on our paper! We highly appreciate your time and consideration!